# Molecular Idiosyncratic Toxicology of Drugs in the Human Liver Compared with Animals: Basic Considerations

**DOI:** 10.3390/ijms24076663

**Published:** 2023-04-03

**Authors:** Rolf Teschke

**Affiliations:** 1Department of Internal Medicine II, Division of Gastroenterology and Hepatology, Klinikum Hanau, D-63450 Hanau, Germany; rolf.teschke@gmx.de; Tel.: +49-6181/21859; Fax: +49-6181/2964211; 2Academic Teaching Hospital of the Medical Faculty, Goethe University Frankfurt/Main, D-60590 Frankfurt am Main, Germany

**Keywords:** idiosyncratic drug induced liver injury (DILI), molecular toxicology, RUCAM, cytochrome P450 (CYP), ROS, hepatic immune system, human leucocyte antigen (HLA) genotypes, hepatic mediators, DILI animal models

## Abstract

Drug induced liver injury (DILI) occurs in patients exposed to drugs at recommended doses that leads to idiosyncratic DILI and provides an excellent human model with well described clinical features, liver injury pattern, and diagnostic criteria, based on patients assessed for causality using RUCAM (Roussel Uclaf Causality Assessment Method) as original method of 1993 or its update of 2016. Overall, 81,856 RUCAM based DILI cases have been published until mid of 2020, allowing now for an analysis of mechanistic issues of the disease. From selected DILI cases with verified diagnosis by using RUCAM, direct evidence was provided for the involvement of the innate and adapted immune system as well as genetic HLA (Human Leucocyte Antigen) genotypes. Direct evidence for a role of hepatic immune systems was substantiated by (1) the detection of anti-CYP (Cytochrome P450) isoforms in the plasma of affected patients, in line with the observation that 65% of the drugs most implicated in DILI are metabolized by a range of CYP isoforms, (2) the DIAIH (drug induced autoimmune hepatitis), a subgroup of idiosyncratic DILI, which is characterized by high RUCAM causality gradings and the detection of plasma antibodies such as positive serum anti-nuclear antibodies (ANA) and anti-smooth muscle antibodies (ASMA), rarely also anti-mitochondrial antibodies (AMA), (3) the effective treatment with glucocorticoids in part of an unselected RUCAM based DILI group, and (4) its rare association with the immune-triggered Stevens-Johnson syndrome (SJS) and toxic epidermal necrolysis (TEN) caused by a small group of drugs. Direct evidence of a genetic basis of idiosyncratic DILI was shown by the association of several HLA genotypes for DILI caused by selected drugs. Finally, animal models of idiosyncratic DILI mimicking human immune and genetic features are not available and further search likely will be unsuccessful. In essence and based on cases of DILI with verified diagnosis using RUCAM for causality evaluation, there is now substantial direct evidence that immune mechanisms and genetics can account for idiosyncratic DILI by many but not all implicated drugs, which may help understand the mechanistic background of the disease and contribute to new approaches of therapy and prevention.

## 1. Introduction

The potential risk of liver injury by the use of conventional drugs [1,2,3] is shared by a variety of other toxicants such as phytochemicals found in regulatory approved herbal drugs or non-approved herbal medicines like Traditional Chinese Medicines (TCM) [4,5]. Humans may also experience liver injury due to aluminum, arsenic, beryllium, cadmium, chromium, cobalt, copper, iron, lead, mercury, molybdenum, nickel, platinum, thallium, titanium, vanadium, zinc, carbon tetrachloride and herbicides [6,7] as well as pesticides [8] amanitin of *Amanita phalloides* [9], aflatoxins [10], plants containing 1,2-unsaturated pyrrolizidine alkaloids (PAs), contaminating food or drinking water [11,12], and alcoholic beverages [13,14,15,16,17,18]. The diagnosis of liver injury due to most of these compounds can be achieved by exposure history, clinical features, diagnostic biomarkers, and after careful exclusion of alternative causes.

With respect to idiosyncratic drug induced liver injury (DILI) in humans, the use of cases with verified diagnosis is mandatory for defining its molecular toxicology [19]. Well known among experts, the diagnosis of this disease is challenging [2,20,21,22,23,24] as well as its presentation of clinical features due to multiple facets of the injury [3] in face of the abundancy of drugs implicated in it [1]. Challenges to be faced and discussed provide broad popularity among scientists, physicians, regulators, and experts in the field [1,2,3,20,21,22,23,24].

In the past, substantial efforts have been undertaken to optimize the diagnosis of idiosyncratic DILI in patients with abnormal liver tests (LTs) observed under a therapy with conventional drugs. Already in 1993, however, straight forward proposals started in France with an international consensus meeting of DILI experts, who inaugurated the Roussel Uclaf Causality Assessment Method (RUCAM) [25,26], updated in 2016 [27]. In line with current knowledge, basic considerations of artificial intelligence (AI) represented the framework of RUCAM with the intention to clarify difficult conditions by simplifying complex processes and providing structured algorithms with specific, scoring key domains [23,24]. Broadly appreciated as well validated method for causality assessment, RUCAM was applied in 81,856 DILI cases reported as single case reports, case series, or study cohorts worldwide published until mid of 2020 [1]. RUCAM was also used in 996 DILI cases described in COVID-19 patients in 2020 and 2021 [28]. The advantages of RUCAM for evaluating the causality in DILI cases was highlighted in summarizing review articles by the group of Lewis et al. [2,20,21,22], the Chinese Society of Hepatology (CSH) together with the Chinese Medical Association (CMA) in their guidelines for the diagnosis and treatment of DILI [29], the DILI consensus guidelines of the Asia Pacific Association of Study of Liver (APASL) [30], another international consensus conference [31], and European DILI registries [32]. With regrets, the LiverTox database does not fulfil requirement of a professional causality assessment due to the note on the LiverTox paradox, based on gaps between promised good data of DILI cases assessed using RUCAM and the reality check that this was not provided [33]. Highly appreciated and analyzed recently [24], RUCAM was also among the topics in a current scientometric study by independent DILI experts from China not affiliated with any known DILI circle [34]. For causality assessment in DILI, there is no place to use electronic versions modified from RUCAM, because all these versions lack valid validation as published for RUCAM [35,36].

In this article, the molecular toxicology of drugs implicated in the human idiosyncratic liver injury and preferentially analyzed by RUCAM based DILI cases was evaluated and to see to whether animal models can be used to shed light into the mechanistic steps leading to this human disease. The current analysis showed convincing direct evidence that human idiosyncratic DILI caused by selected drugs is due to immune systems and genetics, while for the remaining drugs additional studies are required or alternatively, other mechanisms must be invoked. There is increasing awareness that animal models cannot replace studies in affected patients.

## 2. Literature Search and Source

The PubMed database and Google scholar was searched for articles by using the following key terms: idiosyncratic drug induced liver injury (DILI) and RUCAM and mechanistic steps and cytochrome P450 (CYP) and hepatic immune system and DILI animal models. Publications in the English language were preferred and considered following analysis of their suitability. Selected were the most relevant original papers, single case reports, case series, study cohorts, consensus reports, and review articles, with focus on mechanistic results derived from cases of DILI with clear diagnosis verified using RUCAM. There was no restriction of the period of publication date of the reports. Literature search was started on 2 December 2022 and finished on 2 February 2023.

## 3. Definitions

### 3.1. Drugs Most Commonly Implicated in Idiosyncratic DILI and Intrinsic DILI

By convention, DILI is caused by regulatory approved drugs, which may have the potential of causing idiosyncratic or intrinsic liver injury [2,3,6,27,34]. Idiosyncratic liver injury is due to the interaction between the drug used in recommended daily doses and a susceptible individual [6,27], whereby this type of injury can be caused by virtually any conventional drug [7]. As opposed, intrinsic liver injury occurs through drug overdose like paracetamol syn acetaminophen or syn N-acetyl-p-aminophenol (APAP) as the best-known clinical example [6]. Consequently, patients with idiosyncratic DILI used their drugs commonly for some days, weeks, or months, while in the context of an acute intoxication the drug intake is mostly limited to one or two days exceeding the daily allowance of daily dose but can also develop after high cumulative drug doses taken over a longer period [6,7]. To facilitate an overview of drugs most implicated in DILI with verified diagnosis as assessed for causality using RUCAM, a list of selected drugs is provided in alphabetical order (Table 1) [37,38,39,40,41,42,43,44,45,46,47,48,49,50,51,52], with additional details published earlier [53,54].

For most drugs, DILI was verified as diagnosis using RUCAM. Antimetabolites include azathioprine, 6-mercaptopurine, and methotrexate. Data were partially derived from published reports [53,54] that provided case and RUCAM details for each drug implicated in DILI [37,38,39,40,41,42,43,44,45,46,47,48,49,50,51,52]. Of note, some drugs like acetaminophen, amiodarone, anabolic steroids, atorvastatin and other statins, azathioprine/6-mercaptopurine, methotrexate, and valproic acid, which basically cause intrinsic DILI, may also trigger idiosyncratic DILI if used in recommended doses.

The risk of drugs causing human idiosyncratic DILI has been shown in abundant cases published worldwide and assessed for causality using RUCAM [1], conditions confirmed for selected drugs as listed with consideration of some drugs implicated in intrinsic DILI that were not assessed by RUCAM (Table 1) [37,38,39,40,41,42,43,44,45,46,47,48,49,50,51,52]. Therefore, molecular toxicology in human idiosyncratic and intrinsic DILI caused by the use of drugs are best studied in RUCAM based DILI cases.

### 3.2. Criteria of Liver Injury

Analysis of molecular toxicology also requires the mandatory classification of the idiosyncratic DILI, achievable by using LT abnormalities of serum alanine aminotransferase (ALT) activities ≥5 times of the upper limit of normal (ULN) and/or alkaline phosphatase (ALP) activities ≥2 times of the ULN [27] shown in Table 2. Lower thresholds are characteristic of liver adaptation syn tolerance as opposed to the classical real liver injury.

### 3.3. Phenotype Syn Pattern Characteristics of Idiosyncratic DILI

Molecular toxicology in DILI must be studied in the underlying pattern syn phenotype of the liver injury, not relying on liver histology obtained through invasive liver biopsy but rather than on analysis of serum LTs [27]. Thereby, the ratio R is important, obtained by using multiples of ULN of ALT and ALP to be divided as ALT: ALP. The liver injury is hepatocellular if R ≥ 5, the liver injury is cholestatic if R ≤ 2, and the liver injury is mixed if 2 < R < 5. These three injury types are commonly found in DILI cases [1,27,43] but they must be differentiated from drug induced microvesicular steatosis hepatitis due to amiodarone as example [55], drug induced hepatic sinusoidal obstruction syndrome (HSOS) caused by oxaliplatin [56], or drug induced autoimmune hepatitis (DIAIH) [57].

## 4. Principles of Hepatic Drug Uptake, Metabolism, and Elimination

### 4.1. Hepatocellular Drug Uptake

Following oral intake and absorption through the mucosa of the intestinal tract, drugs reach the human liver via the venous portal system. Drugs enter the hepatocytes through a variety of mechanisms, including passive drug diffusion from the blood and active influx via transporters like NTCP (Na+-taurocholate cotransporting polypeptide), OCT (organic cation transporter) and OATP (organic anion transporting polypeptide) [58,59,60,61]. All these processes take place in the sinusoidal plasma membrane of the hepatocyte. Drug uptake in animal studies, is commonly achieved through gastric gavage or via intraperitoneal injection.

### 4.2. Hepatocellular Drug Biotransformation

Within the hepatocytes, drug biotransformation is preferentially achieved by actions of various drug metabolizing enzymes involving, for instance, microsomal cytochrome P450 (CYP) isoforms [59,60], alternatively non-CYP pathways like flavin-containing monooxygenase (FMO), monoamine oxidase (MAO), alcohol dehydrogenase (ADH), acetaldehyde dehydrogenase (ALDH), and aldehyde oxidase (AO) [59,61]. These initial steps are grouped as phase I reactions involving oxidation, reduction, or hydrolysis [59,60,61]. The subsequent pathways proceed via conjugating enzymes and are grouped as phase II reactions [59]. Among these are UDP-glycosyltransferase (UGT), glutathione S-transferase (GST), sulfotransferase (SULT) and N-acetyltransferase (NAT) [59,61].

### 4.3. Drug Elimination

Commonly known as phase III, the elimination of the parent drug or its metabolites occurs after conjugation and release either into the blood in the urine after passing the kidneys or into the biliary system preferentially via the bile canalicular pole of the plasma membrane of the hepatocyte by drug efflux mechanisms through transporters like bile salt export pump (BSEP), BCRP (breast cancer resistance protein), MDR (multidrug resistance protein) and MRP (multidrug resistance-associated protein) [59]. Several hundred drugs can cause DILI [1,62], which makes it difficult to assign for each drug reaction an individual mechanism of liver injury [54,63,64,65,66,67,68,69].

## 5. Overview on Molecular Toxicology in Human Idiosyncratic DILI

Idiosyncratic DILI is well known in patients with genetic predisposition [1,3,27], a characteristic feature not detectable in animals, which makes it difficult to find animal models that reproduce idiosyncratic DILI and could help provide details of its molecular toxicology [19]. Although many theories on mechanistic toxicology in idiosyncratic DILI with a plethora of schematic illustrations have been published, some basic issues remain unresolved in this complex disease due to the variability of partially contradictory proposals [54,70,71]. Not unexpected is the abundancy of promoted molecular mechanisms that relate to drugs with variable chemical structures and variabilities of clinical features. In addition, genetic of patients at risk and the multiplicity of non-parenchymal cells in addition to the hepatocytes and multiple immune cell types contribute to the variability of mechanistic steps. Of special interest at the mechanistic level is the involvement of hepatic CYP isoforms and oxidative stress generating ROS (reactive oxygen species), ferroptosis, hepatic immune system, mediators, gut microbiome, differences of hepatocellular injury and cholestatic injury, and possible animal models.

### 5.1. Basics of Hepatic Microsomal Cytochrome P450 and Its Isoforms

The pathogenesis of idiosyncratic DILI has often been related to CYP dependent drug metabolism, although the respective DILI was not assessed regarding causality for the drug under consideration study, nor was there any verification that the drug was really metabolized by CYP, and the CYP isoform involved in the metabolism of the drug commonly remained unconsidered. Based on published data [64,72,73,74,75,76,77,78,79,80,81,82,83,84,85,86,87,88,89,90,91,92,93,94,95,96,97,98,99,100,101,102,103,104,105], a recent analysis on drugs most implicated in idiosyncratic DILI assessed by RUCAM showed that only a portion of the drugs were substrates of hepatic microsomal CYP (Table 3) [54].

The study cohort assessing the role of CYP consisted of 48 top drugs (Table 3), which have been implicated in triggering of assumed DILI and were derived from the US LiverTox database [32,33,106,107]. It turned out that in at least 28/48 drugs (58.3%), clinical or experimental evidence exists that drug metabolism proceeds via CYP, whereas for the remaining 20 drugs (41.7%) there were negative or missing results of metabolic participation of CYP [54]. Additional analyses revealed that among the various CYP isoforms, CYP 3A4 was the most frequent one involved in the metabolism of drugs implicated in causing DILI [19,69]. Of note, analyses on CYP and CYP isoforms (Table 3) [19,54,69] had been carried out for drugs of published DILI cases tested for causality using RUCAM [53]. Such approach was not feasible in DILI cases of the LiverTox database due to a lack of a robust DILI case causality management [32,33,106,107], a major shortcoming considering that many assumed DILI cases were not related to drug use but can be explained by alternative causes [108].

CYP is primarily localized in the liver and degrades many drugs to harmless metabolites; however, it also has the potential to produce toxic metabolites [12,19,54,99], which may initiate idiosyncratic DILI (Figure 1).

Drugs enter like other substrates the catalytic cytochrome P450 cycle as substrate, shown on top of the cycle. Overall, in the course of mechanistic multi-steps, the drug as substrate leaves the CYP cycle after it is oxidized forming now as a metabolite. In more detail, the first electron is provided to CYP by NADPH + H^+^ via the NADPH CYP reductase, whereby the reduced form of CYP with Fe^2+^ is generated, which finally becomes oxidized again after splitting off the oxidized substrate. CYP becomes then again free for the next substrate to be oxidized (Figure 1) [12,19,69]. Through introduction of molecular oxygen, a multi-compound reactive complex is generated, a process facilitated by inclusion of another electron that commonly is provided through the NADPH CYP reductase or a similar but NADPH independent reductase.

#### 5.1.1. Cytochrome P450, Oxidative Stress and Reactive Oxygen Species

During the process of drug metabolism, the catalytic cycle of CYP is involved in the generation of oxygen split products (Figure 1), providing thereby the basis of hepatic oxidative stress, which proceeds at a low level to generate sufficient ROS that helps sustain physiological functions. Under special conditions in the presence of drugs and likely initiated by genetic predisposition, however, an overproduction of ROS can be assumed, part of which will be used for drug metabolism whereas the remaining ROS may injure the liver if hepatic antioxidant systems are exhausted and lead to idiosyncratic DILI [19,54,59,67,109,110,111]. The injury is triggered by various toxic intermediates such as superoxide radicals, nitric oxide radicals, singlet oxygen, hydrogen peroxide, and peroxyl radicals.

#### 5.1.2. Antibodies against Cytochrome P450 Isoforms and Drugs

The toxic intermediates affect the CYPs as well as the drugs implicated in the liver injury and both may react with the formation of specific antibodies, appreciated as diagnostic biomarkers due to their good specificity and sensitivity [112], and as shown for a few drugs as examples in alphabetical order.

#### 5.1.3. Dihydralazine

Idiosyncratic DILI cases due to the antihypertensive drug Dihydralazine and its analogue Hydralazine were rarely described in the literature [27,113,114]. Whereas Hydralazine is found at range 33 among the drugs most implicated in DILI, Dihydralazine is not listed (Table 3) [19,53,54]. Hydralazine interacts with the CYP 1A2 isoform by inhibiting its metabolic property, but it is not metabolized itself by CYP 1A2 or any other CYP isoform [115], with the consequence that anti-CYP antibodies in cases of DILI by Hydralazine are not to be expected. Instead, Dihydralazine may cause idiosyncratic DILI with verified causality for this drug using RUCAM [27], is metabolized by CYP 1A2 isoform [90], and may trigger anti-CYP 1A2 antibodies [116], detectable in the blood of an affected patient [90]. In addition, covalent binding of Dihydralazine metabolites to microsomes from rat and human livers was described [116], shown upon incubation of microsomes with NADPH [116]. These metabolites concomitantly reacted with heme as evidenced by destruction of heme, formation of 445-nm light-absorbing complexes, and covalent binding of heme to apoprotein. Formation of these metabolites was shown by NADPH dependence, induction by beta-naphthoflavone, and immunoinhibition by anti-CYP 1A antibodies to be mediated by CYP 1A. Finally, the metabolites appeared to bind to CYP 1A2, which produced them. Summarizing these events to be classified as autoimmune reaction: CYP 1A2 metabolizes Dihydralazine with the formation of reactive metabolites, which then bind to it, thereby forming a neoantigen that triggers an immune response characterized by autoantibodies against CYP 1A2. This excellent initial study of a drug induced immunoallergic hepatitis was published in 1990 [90], and the case was therefore not assessed using RUCAM, which was published only in 1993 [25]. In other case studies, however, DILI by Dihydralazine was verified as diagnosis using RUCAM [27].

#### 5.1.4. Halothane

Halothane undergoes oxidative metabolism via CYP 2E1 [74,117] and is listed at range 18 among the drugs most implicated in DILI (Table 3) [19,53,54] with established diagnosis using RUCAM in 15 cases [38]. Toxic liver injury due to the volatile anesthetic Halothane is frequently associated with the appearance of serum anti-CYP 2E1 antibodies [19,65,111,118,119,120,121,122], verified by clinical studies that 25/56 (45%) patients diagnosed with Halothane hepatitis have autoantibodies, which react with human CYP 2E1 that was purified from a baculovirus expression system [122]. The autoantibodies inhibited the activity of CYP 2E1 and appeared to be directed against mainly conformational epitopes. In addition, because CYP 2E1 became trifluoroacetylated when it oxidatively metabolized Halothane, it is possible that the covalently altered form of CYP 2E1 may be able to bypass the immunologic tolerance that normally exists against CYP 2E1. Though CYP 2E1 is the predominant isoform in human oxidative Halothane metabolism, CYP 2A6 plays a contributory role [117]. In addition, an in vitro reductive pathway exists for Human halothane metabolism, catalyzed by CYP 2A6 and 3A4 [123].

#### 5.1.5. Isoniazid (INH)

Isoniazid (INH)-induced liver injury ranks at place #13 among the drugs most implicated in idiosyncratic DILI (Table 3) with 19 DILI cases assessed for causality using RUCAM to verify the diagnosis [37,38,39]. Metabolism of INH involves CYP (Table 3) with its preferred isoform CYP 2E1 [75] and CYP 3A4 [124], perhaps also CYP 2D9 [125]. For evaluating antibodies against these CYPs, patients with acute liver failure due to INH use were included in the study, but confirmation of the DILI diagnosis by RUCAM was lacking for the cohort [125]. In the sera of 15/19 patients, antibodies of CYP isoforms were found: 11 sera had anti-CYP 2E1 antibodies, 14 sera had antibodies against CYP 2E1 modified by INH, 14 sera had anti-CYP 3A4 antibodies, and 10 sera had anti-CYP 2C9 antibodies [125].

#### 5.1.6. Sevoflurane

The volatile anesthetic Sevoflurane is a rare cause of DILI, not listed among the top drugs implicated in DILI (Table 1 and Table 2) [53,54]. CYP 2E1 is the principle if not sole isoform catalyzing the metabolism of sevoflurane, assessed by Disulfiram, an inhibitor of CYP 2E1 [126]. The clinical diagnosis of liver injury by Sevoflurane was verified by using RUCAM in four patients with serum anti-CYP 2E1 antibodies leading to causality gradings of highly probable [127].

#### 5.1.7. Tienilic Acid

Tienilic acid is now off the market, which explains the rarity of case reports on its potential to cause autoimmune DILI and the missing listing among the drugs most involved in DILI (Table 1 and Table 3) [53,54]. Respective RUCAM based DILI case reports are not available that would allow a robust clinical feature description except that Tienilic acid is known for causing a drug-induced autoimmune hepatitis (DIAIH) [128]. CYP 2C9 is involved in the metabolism of Tienilic acid, representing a target for Tienilic acid-reactive metabolites and for autoantibodies. To study the relationship between drug metabolism and the mechanistic steps leading to this DIAIH, the specificity of anti-LKM2 (anti-liver and -kidney microsomal type 2) autoantibodies found in the sera of patients toward CYP 2C9 was determined, and the antigenic sites on CYP 2C9 were localized. By constructing several deletion mutants derived from CYP 2C9 cDNA and by probing the corresponding proteins with different anti-LKM2 sera, three regions were defined: amino acids 314–322, 345–356, and 439–455 [128]. They interacted to form a major conformational autoantibody binding site. This binding site was immunoreactive with 100% of sera and allowed removal of the entire reactivity of the sera tested by immunoblotting.

## 6. Hepatic Immune System

Immunological aspects of idiosyncratic DILI have attracted much interest among clinical physicians and theoretical scientists, who presented their views mostly as narrative proposals in multiple publications [57,70,129]. Robust evidence based on real cases assessed using RUCAM was rarely provided, and proposals remained speculative and controversial. As expected from the multiplicity of drugs and interacting processes implicated in the disease, their attempts lacked uniformity and provided a multifaceted picture, based at best on circumstantial evidence only [54,70]. To overcome these issues, more clarification may be obtained when future immunological data are derived from cases of idiosyncratic DILI with verified causality using the updated RUCAM. A low case number is certainly not a problem, considering that worldwide 81,856 DILI cases have been published until mid-2020, all assessed for causality by using RUCAM [1].

### 6.1. Direct Evidence

Direct evidence for a role of the hepatic immune system in cohorts of specific idiosyncratic DILI caused by selected drugs was provided by cases with verified causality using RUCAM to ensure that the DILI was not attributed to alternative causes commonly observed in DILI cohorts [108]. For instance, serum antibodies against the CYP 2E1 isoform have been described in RUCAM based cases of idiosyncratic DILI caused by Sevoflurane as a typical well documented example [127]. In various studies, other drugs have been shown to cause serum antibodies against CYP isoforms, but respective DILI cases with diagnoses confirmed by RUCAM were not presented. Similarly, direct evidence for the involvement of the hepatic immune system in a DILI subgroup was provided by studies on RUCAM based cases of DIAIH caused by some drugs as examples [130]: Antimicrobials [131,132], Atorvastatin [130], Augmentin [131], Ceftriaxone [131], Diclofenac [133], Direct oral anticoagulants [134], Hydralazine [133], Infliximab [135,136], Isoniazid [133], Ketoprofen [131], Minocycline [133], Methyldopa [133], Nimesulide [131], Nitrofurantoin [133,135,137,138], Non-steroidal anti-inflammatory drugs [131,134,137,139], Sorafenib [130], and Statins [132,134,138]. The studies discussed above provided a clear differentiation of the classical genuine AIH from DIAIH by using scores of the simplified AIH scale for assessing the AIH [140] and applying the scores of RUCAM [25,27] for evaluating DIAIH [130]. Apart from triggering DIAIH, part of these drugs may also cause common DILI without signs of autoimmunity, as noted by one study [135] and confirming previous statements [70]. For many of the published DIAIH cases, positive serum antinuclear antibodies (ANA) and anti-smooth muscle antibodies (ASMA) were detected, rarely also ant mitochondrial antibodies (AMA) [131,132,133,134,135,136,137,138,139]. Finally, and in support of the immune involvement, DIAIH responds well to the immune modulatory action of glucocorticoid treatment without relapse after treatment cessation, whereas relapse in genuine AIP is common in and characteristic for this disease [130,134]. Interesting is the fact that that glucocorticoids are only partially effective in treating unselected idiosyncratic DILI caused by various drugs as a whole DILI cohort, suggesting that not all DILI cases are triggered by immune mechanisms [141] in line with previous proposals [70].

Direct evidence for an involvement of the immune system in idiosyncratic DILI was also provided by its rare association with the immune-triggered Stevens-Johnson syndrome (SJS) and toxic epidermal necrolysis (TEN) caused by a small group of drugs [142]. Causality of idiosyncratic DILI was evaluated by RUCAM and of SJS/TEN using the Algorithm for Drug Causality for Epidermal Necrolysis, which was highly probable or probable in all cases.

Direct evidence for a role of the innate and adaptive immune system in idiosyncratic DILI with RUCAM based verification of the diagnosis is increasingly observed. The initiation of an immune response requires activation of antigen presenting cells (APCs) by molecules such as danger-associated molecular pattern molecules (DAMPs) [70]. Direct evidence for the involvement of the innate immune system in the idiosyncratic DILI was shown with causative drugs such as Diclofenac, Indomethacin, Levofloxacin, and Phencoumon by studies of monocyte-derived hepatocyte-like cells in DILI cases assessed by RUCAM [143], in line with considerations that monocytes are part of the innate immune system [144,145,146,147]. In short, hepatic monocytes are commonly derived from bone marrow progenitors, released into the blood before they enter the liver, where they differentiate into liver resident macrophages such as Kupffer cells (KCs) and infiltrating monocyte-derived macrophages (MoMF), allowing for crosstalk with liver monocytes within the liver with intensive exchange of inflammatory mediators [148]. Using commercially available kits, they are principally measurable as circulatory mediators such as the cytokines IL-22, IL-22 binding protein (IL-22BP), IL-6, IL-10, IL 12p70, IL-17A, IL-23, IP-10, or chemokines like CD206 and sCD163 in the plasma of patients with the diagnosis of DILI caused by anti-tuberculosis drugs and verified by the prospective use of the updated RUCAM that provided high causality gradings [149]. In addition, the parameters IP-10 and sCD163 can be used even as risk factors of future cases of this DILI entity. Direct evidence that idiosyncratic DILI is partly mediated by the adaptive immune system can be traced back to the fact that the DILI caused by a few drugs is associated with specific HLA genotypes [70], found along with GWAS (Genome wide association study) in RUCAM based cases of DILI due to anti-tuberculosis drugs [150], Flucloxacillin [151], and Amoxicillin-clavulanate [152], but with lack of data reproducibility using this drug combination as reported in another RUCAM based report [153]. An assumed HLA association of liver injury by Amoxicillin-clavulanate has been proposed already in 1999, but results remained vague because cases were not assessed using RUCAM. Of clinical significance, Amoxicillin clavulanate was at rank #1 and Flucloxacillin at rank #2 among the drugs most implicated in RUCAM based DILI worldwide (Table 3), and regarding mechanistic steps, drug metabolism via CYP is not essential for HLA mediated DILI because Amoxicillin clavulanate is not metabolized by CYP (Table 3) [72] as opposed to Flucloxacillin that is metabolized (Table 3) [73].

Finally, direct evidence of an immune involvement in an idiosyncratic DILI was recently provided in a highly appreciated prospective study in humans by urine metabolomics and microbiome analyses, which revealed the mechanism of DILI caused by anti-tuberculosis drugs with verified diagnosis, as assessed for causality using the updated RUCAM [154].

### 6.2. Lack of Valid Evidence

Gaps are obvious in a few areas of the hepatic immune system related to idiosyncratic DILI. More specifically, clear initial mechanistic immune-based clues triggering the idiosyncratic liver injury are poorly understood, speculative, and not based on evidence. In this context, there is also uncertainty how inflammatory mediators interact with parenchymal and non-parenchymal cells of the liver and how immunological processes affect the liver histology, function and integrity of liver mitochondria and bile salt export pumps (BSEP) [70]. There are no evidence-based data on the possible role of ferroptosis and pyroptosis [141] for the hepatic immune system in idiosyncratic DILI with established diagnosis using RUCAM.

## 7. Animal Models

There is increasing awareness that the human model of idiosyncratic DILI rather than any animal model is ideal and appropriate to search for mechanistic clues leading to this specific disease [1], provided data are derived from cases assessed for causality using RUCAM [25,26], best now as its update, which among other requirements also strongly asks for mandatory rather than mere optional exclusion of infections by HEV (hepatitis E virus) [27]. Neglecting RUCAM based DILI cases for evaluating mechanistic steps of idiosyncratic DILI is viewed a major scientific, clinical, and theoretical flaw, not helpful for the DILI community. In fact, many discussions and proposals of pathogenetic aspects focused on data derived from studies using animals, which do not mimic human features such as their specific immune system or HLA genotypes. For example, animal related topics included broad discussions on experimental methods using predictive models, in vitro models, and in vivo models [70,155]. Proposals were critically assessed and led to the correct conclusion that current in vitro and in vivo systems were still not adequate enough to understand the pathogenetic mechanisms of this complex condition of idiosyncratic DILI. To deepen the discussion, it is certainly tempting to approach mechanistic steps of human idiosyncratic DILI by using, for instance, in vitro hepatocytes derived from the rat liver to study cytotoxic effects of Bupropion, a broadly used antidepressant drug for smoke cessation [156]. Bupropion led in vitro to an increased production of ROS, associated with a depletion of intracellular glutathione, elevation of lipid peroxides, and mitochondrial collapse. Such changes were not described in patients with RUCAM based DILI by this drug, not allowing for translation of these experimental data to real DILI in humans. Similar conclusions of cautionary may be drawn from other experimental studies using drugs like Citalopram [157] and Rizatriptan [158]. Finally, whether Zebrafish is an appropriate experimental study tool for idiosyncratic DILI comparable with RUCAM based human cases remains to be established [159].

## 8. Conclusions

Idiosyncratic DILI is a classical human disease of the liver, well defined by clinical features reported in 81,566 cases with verified diagnosis by using RUCAM to establish causality. Data derived from the current mechanistic analysis help understand pathogenetic principles of the hepatic immune systems leading to DILI and may contribute to search for new therapeutic approaches urgently needed for DILI patients, providing therefore substantial advantages for future biomedical applications. With respect to genetic data retrieved from RUCAM based DILI reports, there is increasing awareness that genetic variation represents a risk factor for the development of DILI, which will require a more critical consideration by the physician before initiation of a treatment with a new drug. New challenges emerged from this analysis because a portion of this disease is not related to these pathways, illustrating its inhomogeneity and explaining the variability of treatment efficacy attempted by various therapeutic approaches. Using the perfect RUCAM based DILI cohort further studies are now needed to close the gaps of mechanistic features among the large DILI group with different immunological and genetic background susceptible to liver injury caused by more than 1000 drugs. To study this, using cohorts consisting of patients with RUCAM based DILI are the optimal approach whereas animal models are not the ideal tool due to lack of mimicking individual inherited characteristic features of affected humans.

## Figures and Tables

**Figure 1 ijms-24-06663-f001:**
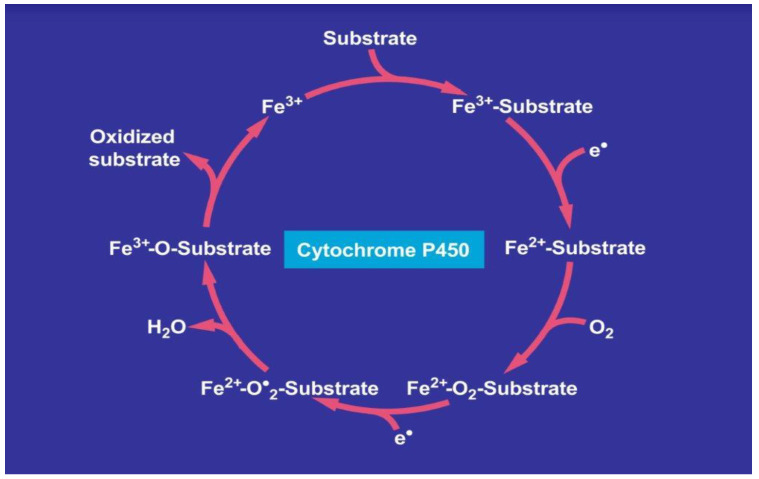
**Catalytic CYP cycle of hepatic microsomal drug metabolism.** Cytochrome P450 stands for its various isoforms. The term “P450” was proposed to describe a “pigment” with an absorption maximum at 450 nm with the ferrous-carbon monoxide complex of CYP in rat liver microsomes. The figure was adapted from recent open access reports [12,69].

**Table 1 ijms-24-06663-t001:** Selected drugs most implicated in idiosyncratic and intrinsic DILI.

Selected Drugs Most Commonly Implicated in DILI	RUCAM Used	References First Author
** *Idiosyncratic liver injury* **		
Allopurinol	YES	Douros 2014 [37]
Amoxicillin-clavulanate	YES	Björnsson 2005 [38], Andrade 2005 [39], Andrade 2006 [40], García-Cortés 2008 [41]
Carbamazepine	YES	Björnsson 2005 [38], Andrade 2005 [39]
Chlorpromazine	YES	Björnsson 2005 [38], Zhu 2016 [42]
Contraceptives	YES	Douros 2014 [37], Björnsson 2005 [38], Wai 2006 [44]
Diclofenac	YES	Douros 2014 [37], Björnsson 2005 [38], Andrade 2005 [39], Andrade 2006 [40]
Dihydralazine	YES	Douros 2014 [37]
Disulfiram	YES	Björnsson 2005 [38]
Erythromycin	YES	Björnsson 2005 [38], Andrade 2005 [39]
Flucloxacilllin	YES	Douros 2014 [37], Björnsson 2005 [38]
Flupirtine	YES	Douros 2014 [37]
Flutamide	YES	Andrade 2005 [39]
Halothane	YES	Björnsson 2005 [38]
Ibuprofen	YES	Douros 2014 [37], Björnsson 2005 [38], Andrade 2005 [39], Zhu 2016 [42]
Infliximab	YES	Douros 2014 [37]
Interferon alpha/ Peginterferon	YES	Rathi 2017 [43]
Interferon beta	YES	Douros 2014 [37]
Isoniazid	YES	Douros 2014 [37], Björnsson 2005 [38], Andrade 2005 [39]
Ketoconazole	YES	Zhu 2016 [42]
Natriumaurothiolate	YES	Björnsson 2005 [38]
Nimesulide	YES	Andrade 2005 [39], Zhu 2016 [42], Rathi 2017 [43]
Nitrofurantoin	YES	Andrade 2005 [39], Zhu 2016 [42]
Phenytoin	YES	Andrade 2006 [40]
Propylthiouracil	YES	Zhu 2016 [42]
Pyrazinamide	YES	Douros 2014 [37]
Rifampicin	YES	Douros 2014 [37], Björnsson 2005 [38]
Sulfamethoxazole/Trimethoprim	YES	Björnsson 2005 [38]
Sulfazalazine	YES	Björnsson 2005 [38]
Sulindac	YES	Douros 2014 [37]
Ticlopidine	YES	Björnsson 2005 [38], Andrade 2005 [39], Wai 2006 [44]
** *Intrinsic liver injury* **		
Acetaminophen	YES	Teschke 2020 [45], Teschke 2016 [46]
Amiodarone	YES	Douros 2014 [37]
Anabolic steroids	YES	Zhu 2016 [42]
Atorvastatin and other statins	YES	Douros 2014 [37], Björnsson 2005 [38], Andrade 2005 [40], Zhu 2016 [42]
Antimetabolites	YES	Douros 2014 [37], Björnsson 2005 [38], Andrade 2005 [39]
Cholestyramine	NO	Singhal 2014 [47]
Cyclosporine	NO	Kassianides 1990 [48]
HAART drugs	NO	Inductivo-Yu 2008 [49]
Heparins	NO	Bosco 2019 [50]
Nicotinic acid	NO	Clementz 1987 [51]
Tacrine	NO	Blackard 1998 [52]
Valproic acid	YES	Douros 2014 [37], Andrade 2005 [39], Andrade 2006 [40], Zhu 2006 [42]

Abbreviations: DILI, Drug induced liver injury; HAART, Highly active antiviral therapy; RUCAM, Roussel Uclaf Causality Assessment Method. Table was modified and updated from a previous open access report [53].

**Table 2 ijms-24-06663-t002:** Thresholds of serum ALT and ALP activities in patients with liver injury and liver adaptation.

Description	Thresholds of Liver Tests	Characteristic Features
**Idiosyncratic liver injury**	ALT ≥ 5 times of ULN and/or ALP ≥ 2 times of ULN	Develops at low doses of a drug Signs of liver injury found in histology Cessation of drug use is mandatory and immediate Worsening if drug use is continued Most drugs cause idiosyncratic DILI Risk of acute liver failure
**Intrinsic liver** **injury**	ALT ≥ 5 times of ULN and/or ALP ≥ 2 times of ULN	Develops with overdosed drugs Signs of liver injury found in histology Cessation of use is mandatory and immediate Caused by a few drugs Risk of acute liver failure
**Liver adaptation**	ALT ≤ 5 times of ULN and/or ALP ≤ 2 times of ULN	Develops at low doses of a drug Presumably most drugs have the potency of causing rare but clinically not apparent liver adaptation No signs of liver injury in histology Normalization or stabilization of liver tests is commonly observed whether the drug use is stopped or continued

Abbreviations: ALT, Alanine aminotransferase; ALP. Alkaline phosphatase; ULN, Upper limit of the normal range. This table is derived from a previous open access journal [6,27].

**Table 3 ijms-24-06663-t003:** CYP involvement in DILI by various drugs as assessed in RUCAM based cases.

Drugs Most Commonly Implicated in Causing Idiosyncratic DILI	DILI Cases Assessed Using RUCAM (n)	Substrates of CYP	References First Author
1. Amoxicillin-clavulanate	333	-	Hautekeete 1999 [72]
2. Flucloxacilllin	130	CYP 3A4	Dekker 2019 [73]
3. Atorvastatin	50	CYP 3A/5	Zanger 2013 [74]
4. Disulfiram	48	CYP 2E1	Hopley 2006 [75]
5. Diclofenac	46	CYP 2C8	Zanger 2013 [74]
6. Simvastatin	41	CYP 3A4/5	Fatunde 2010 [76]
7. Carbamazepine	38	CYP 3A4/5	Zanger 2013 [74]
8. Ibuprofen	37	CYP 2C8/9	Hopley 2006 [75]
9. Erythromycin	27	CYP 3A4	Hopley 2006 [75]
10. Anabolic steroids	26	CYP 2C19	Yamazaki 1997 [77]
11. Phenytoin	22	CYP 2C9	Hopley 2006 [75]
12. Sulfamethoxazole/Trimethoprim	21	CYP 2C9	Hopley 2006 [75]
13. Isoniazid	19	CYP 2E1	Hopley 2006 [75]
14. Ticlopidine	19	CYP 2C10	Hopley 2006 [75]
15. Azathioprine/6-Mercaptopurine	17	-	Johansson 2011 [64]
16. Contraceptives	17	CYP 3A4	Scott 2008 [78]
17. Flutamide	17	CYP 1A2	Zanger 2013 [74]
18. Halothane	15	CYP 2E1	Zanger 2013 [74]
19. Nimesulide	13	CYP 2C9	Yu 2014 [79]
20. Valproic acid	13	CYP + 2C9	Kiang 2006 [80]
21. Chlorpromazine	11	CYP 2D6	Hopley 2006 [75]
22. Nitrofurantoin	11	-	Wang 2008 [81]
23. Methotrexate	8	-	Donehower 2008 [82]
24. Rifampicin	7	-	Acocella 1983 [83]
25. Sulfasalazine	7	-	Das 1997 [84]
26. Pyrazinamide	6	-	Shih 2013 [85]
27. Sodium Aurothiomalate	5	-	Björnsson 2005 [38]
28. Sulindac	5	CYP 1A2	Brunell 2011 [86]
29. Amiodarone	4	CYP 3A4	Hopley 2006 [75]
30. Interferon beta	3	-	Bertz 1997 [87]
31. Propylthiouracil	2	CYP NA	Heidari 2015 [88]
32. Allopurinol	1	-	Turnheim 1999 [89]
33. Hydralazine	1	-	Bourdi 1990 [90]
34. Infliximab	1	-	LiverTox 2017 [91]
35. Interferon alpha/Peginterferon	1	-	Okuno 1990 [92]
36. Ketoconazole	1	-	Kim 2017 [93]
37. Busulfan	0	-	Myers 2017 [94]
38. Dantrolene	0	-	Amano 2018 [95]
39. Didanosine	0	-	Andrade 2011 [96]
40. Efavirenz	0	CYP 2B6	Desta 2007 [97]
41. Floxuridine	0	-	Landowski 2005 [98]
42. Methyldopa	0	CYP NA	Dybing 1976 [99]
43. Minocycline	0	-	Nelis 1982 [100]
44. Telithromycin	0	CYP 3A4	Shi 2005 [101]
45. Nevirapine	0	CYP 3A4	Erickson 1999 [102]
46. Quinidine	0	CYP 3A4	Nielsen 1999 [103]
47. Sulfonamides	0	CYP NA	Back 1988 [104]
48. Thioguanine	0	-	Choughule 2014 [105]

Listed are the top ranking 48 drugs implicated in causing 3312 idiosyncratic DILI cases with verified causality using RUCAM for 36 drugs and without verification for 10 drugs and references provided earlier [53] as well as in Table 1. Note: some of the listed drugs may also cause intrinsic DILI if used in acute overdose or during a long treatment duration. Abbreviations: CYP, Cytochrome P450; DILI, Drug induced liver injury; NA, not available. RUCAM, Roussel Uclaf Causality Assessment Method. Table was modified and taken from earlier open access reports [19,53,54].

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
