# Peer review of "Molecular Idiosyncratic Toxicology of Drugs in the Human Liver Compared with Animals: Basic Considerations"

_ijms, 2023, doi:10.3390/ijms24076663_

Round 1

Reviewer 1 Report

This manuscript deals with "Molecular idiosyncratic toxicology of drugs in the human liver compared with animals: Basic considerations". This article claims immune mechanisms and genetics can account for idiosyncratic DILI. This result is very interesting and therefore, I suggest a minor correction and require a detailed clarification. Correction to be addressed by the authors as follows: The abstract is not well organized, where the sentences are incomplete and no continuity is there. It would be feasible, if include the significance of the current study in the abstract. A brief description of how the authors selected information from the literature in the databases, as well as what time period they searched for, is missing. Authors should justify and expand the information on the advantages of this study for biomedical applications. Authors should specify the main experimental conditions used on the evidences from the literature. Where they briefly describe the most important data reported in the literature in a homogeneous manner and sequence reinforcing the relevance of of this approach. Authors should discuss the effect of idiosyncratic DILI on mitochondria and also antioxidants levels  Please add below studies to your manuscript in discussion section using below manuscripts:

DOI: 10.1055/s-0042-123034

DOI: 10.1055/s-0042-110178

DOI: 10.1007/s12272-016-0766-0

Conclusions should reaffirm the fundamental contribution of this paper.

Author Response

Reviewer 1

Thank you for providing constructive points.

  1. Abstract was re-written, significance was included.
  2. A brief description was included under 2. Literature search and source.
  3. Advantages and evidences were included in 8. Conclusions and in abstract.
  4. Requested references were included and discussed under 7. Animal models.
  5. The English was improved.

Reviewer 2 Report

Dr. Teschke's review deals in its entirety with one of the most discussed topics of recent years, drug-induced liver injury.

Despite the fact that there are many articles on the same subject in the literature, I consider this manuscript to be one of the most comprehensive and complete on the subject, in particular for its well-described and summarised analysis of toxicological idiosyncrasy in humans, the involvement of the immune system, and the need for further investigation of the role that the enzymes of the hepatic CYP450 family play in hepatotoxicity from drugs and derivatives.

The manuscript needs only minor editing.

Page 2 lines 51-56: where possible, avoid repetition and redundancy. In this section the acronym DILI is repeated several times in successive sentences, I think the use of different expressions/words could be considered.

Page 2 line 95: cit. 'lowing analysis of their for suitability'. is this a misspelling or typo error?

Author Response

 Reviewer 2:

Thank you for appreciation of my work.

  1. Page 2 lines 51-56, modification was done, shown in color.
  2. Page 2 line 95, typo was corrected, shown in color.